# Altered functional connectivity during speech perception in congenital amusia

Kyle Jasmin[1,2]*, Frederic Dick[1,3], Lauren Stewart[4], Adam Taylor Tierney[1]

[1]Department of Psychological Sciences, Birkbeck University of London, London, United Kingdom; [2]UCL Institute of Cognitive Neuroscience, University College London, London, United Kingdom; [3]Department of Experimental Psychology, University College London, London, United Kingdom; [4]Department of Psychology, Goldsmiths University of London, London, United Kingdom

**Abstract** Individuals with congenital amusia have a lifelong history of unreliable pitch processing. Accordingly, they downweight pitch cues during speech perception and instead rely on other dimensions such as duration. We investigated the neural basis for this strategy. During fMRI, individuals with amusia (N = 15) and controls (N = 15) read sentences where a comma indicated a grammatical phrase boundary. They then heard two sentences spoken that differed only in pitch and/or duration cues and selected the best match for the written sentence. Prominent reductions in functional connectivity were detected in the amusia group between left prefrontal language-related regions and right hemisphere pitch-related regions, which reflected the between-group differences in cue weights in the same groups of listeners. Connectivity differences between these regions were not present during a control task. Our results indicate that the reliability of perceptual dimensions is linked with functional connectivity between frontal and perceptual regions and suggest a compensatory mechanism.

*For correspondence:
kyle.jasmin.11@ucl.ac.uk

Competing interests: The authors declare that no competing interests exist.

## Introduction

Congenital amusia is a rare condition characterized by impaired perception of and memory for pitch (*Peretz et al., 2002*). Although congenital amusia presents as an auditory condition, auditory cortical responses are normal (*Moreau et al., 2013*; *Norman-Haignere et al., 2016*), as is subcortical encoding of pitch (*Liu et al., 2015b*). The dominant view of amusia's neural basis is that connectivity between right inferior frontal cortex and right auditory cortex is impaired, resulting in impaired conscious access to pitch information for guiding behavior (*Hyde et al., 2011*; *Albouy et al., 2013*; *Leveque et al., 2016*; *Zendel et al., 2015*; see *Peretz, 2016* for review). While congenital amusia is believed to be innate, there is evidence that recovery is possible through training (*Whiteford and Oxenham, 2018*).

Although pitch is usually associated with music, it is also important for cueing categories in spoken language (*de Pijper and Sanderman, 1994*; *Streeter, 1978*) and conveying emotion in speech (*Frick, 1985*). In highly controlled laboratory tasks in which speech perception judgments must be made based on pitch alone, only minor deficits have been observed in amusia (*Liu et al., 2015a*; *Patel et al., 2008*). In naturalistic speech perception contexts, people with amusia rarely report any difficulties (*Liu et al., 2010*). This may be because, in natural speech, pitch variation tends to co-occur with variation in other acoustic dimensions, such as duration and amplitude. Our lab has shown that in such cases where multiple redundant cues are available, English-speaking individuals with amusia tend to rely less on pitch than non-amusic controls, suggesting they may calibrate their perception by down-weighting the cues that are less reliable for them (*Jasmin et al., 2020a*). As for emotional prosody in speech, individuals with amusia can recognize emotions in spoken sentences,

**eLife digest** Spoken language is colored by fluctuations in pitch and rhythm. Rather than speaking in a flat monotone, we allow our sentences to rise and fall. We vary the length of syllables, drawing out some, and shortening others. These fluctuations, known as prosody, add emotion to speech and denote punctuation. In written language, we use a comma or a period to signal a boundary between phrases. In speech, we use changes in pitch – how deep or sharp a voice sounds – or in the length of syllables.

Having more than one type of cue that can signal emotion or transitions between sentences has a number of advantages. It means that people can understand each other even when factors such as background noise obscure one set of cues. It also means that people with impaired sound perception can still understand speech. Those with a condition called congenital amusia, for example, struggle to perceive pitch, but they can compensate for this difficulty by placing greater emphasis on other aspects of speech.

Jasmin et al. showed how the brain achieves this by comparing the brain activities of people with and without amusia. Participants were asked to read sentences on a screen where a comma indicated a boundary between two phrases. They then heard two spoken sentences, and had to choose the one that matched the written sentence. The spoken sentences used changes in pitch and/or syllable duration to signal the position of the comma. This provided listeners with the information needed to distinguish between "after John runs the race, ..." and "after John runs, the race...", for example.

When two brain regions communicate, they tend to increase their activity at around the same time. The brain regions are then said to show functional connectivity. Jasmin et al. found that compared to healthy volunteers, people with amusia showed less functional connectivity between left hemisphere brain regions that process language and right hemisphere regions that process pitch. In other words, because pitch is a less reliable source of information for people with amusia, they recruit pitch-related brain regions less when processing speech.

These results add to our understanding of how brains compensate for impaired perception. This may be useful for understanding the neural basis of compensation in other clinical conditions. It could also help us design bespoke hearing aids or other communication devices, such as computer programs that convert text into speech. Such programs could tailor the pitch and rhythm characteristics of the speech they produce to suit the perception of individual users.

but not in short samples such as isolated vowels (*Pralus et al., 2019*), or when speech has been filtered to remove high-frequency non-pitch cues (*Lolli et al., 2015*).

It is unknown how decreased reliance on a particular acoustic cue during speech perception (such as pitch cues in amusia) is reflected in the brain. Previous neural studies of cue integration have focused on integration of multiple modalities, for example the 'weighted connections' model of multisensory integration. In this model, the relative reliability of the modalities involved with perception of a stimulus is related to differential connectivity strength (*Beauchamp et al., 2010*; *Rohe and Noppeney, 2018*). For example, when participants simultaneously view and feel touches to the hand, and reliability of visual and tactile perception is manipulated experimentally via introduction of noise, connection strength (effective connectivity measured with functional MRI and structural equation modeling) between unimodal and multimodal sensory areas adjusts accordingly. More concretely, when visual information is degraded, the connection strength between lateral occipital cortex (a visual area) and intraparietal sulcus (a multimodal area) decreases, and when tactile perception is made noisier, connection strength between secondary somatosensory cortex and intraparietal sulcus becomes weaker (*Beauchamp et al., 2010*). Similarly, effective connectivity between the (multimodal) superior temporal sulcus (STS) and visual and auditory areas has shown similar modulations during processing of audiovisual speech: connection strength between auditory cortex and the STS is weaker when noise has been introduced to the auditory speech, and conversely connection strength between visual cortex and STS is weaker if visual noise is introduced (*Nath and Beauchamp, 2011*).

Just as connectivity differences have been shown to reflect the precision of different sensory modalities during multi*sensory* integration, an analogous phenomenon may be at work within a single modality during multi*dimensional* integration. As mentioned, the acoustic speech signal carries multiple co-occurring acoustic dimensions (e.g. roughly described as voice pitch, duration, and amplitude), which often provide redundant cues to disambiguate a linguistic category (*Patel, 2014*; *Winter, 2014*; *Jasmin et al., 2020a*). Individuals with typical pitch perception have learned through a lifetime of experience with speech acoustics that vocal pitch is a useful and reliable cue. By contrast, individuals with amusia, who have unreliable perception of and memory for pitch (analogous to the 'noise' introduced in the multisensory integration studies cited above), would have learned that, for them, pitch is not a reliable cue for processing spoken language. Thus, by analogy to the multisensory weighting results described above, we hypothesize that amusics may exhibit decreased connectivity between language regions and pitch-related areas during speech processing.

The neural foundations of perceptual weighting in speech have thus far not been investigated in atypical individuals. Indeed, only one previous functional neuroimaging study has examined the neural processing of spoken material in people with amusia. In this study, no group differences were detected in task-related activation or functional connectivity during processing of speech (whereas group differences were observed during processing of tones; *Albouy et al., 2019*). However, the connectivity analyses in this study focused on the silent retention interval in a task in which participants needed to maintain phonemic and not pitch-related information in memory; the analyses also used broader bilateral ROIs within networks associated with language processing. It remains an open question how functional connectivity in amusic and non-amusic participants may differ during speech encoding in pitch-related language tasks within regions of interest selected with a whole-brain data-driven approach.

To determine whether the relative reliability of auditory dimensions in speech perception is reflected in functional connectivity, we used functional magnetic resonance imaging to scan 15 individuals with amusia and 15 controls. Participants matched spoken sentences with visually presented ones on the basis of the position of intonational phrase boundaries. These intonation changes were conveyed differently, in three conditions: Pitch-Informative (where only pitch cues could be used to make the judgment), Duration-Informative (where only duration cues could be used) or Both-Informative (both pitch and duration cues could be used; *Jasmin et al., 2020a*; *Jasmin et al., 2020b*). Functional connectivity was then examined using a data-driven approach that allowed us to identify the largest group differences, without the need for regions of interest to be selected *a priori*. The benefit of this approach is that any set of regions could emerge, not only ones reported in previous literature. Crucially, task performance was matched between the groups (based on prior behavioural testing; *Jasmin et al., 2020a*), ensuring that any neural differences did not simply represent an inability to perform the task. Finally, functional connectivity between these areas was analyzed with respect to prosodic cue weights obtained outside the scanner, and also compared to functional connectivity calculated from different scanning runs with a passive listening task.

## Results

### In-scanner behavior

On each trial, participants read one visually presented text sentence, then heard two auditory versions of the sentence, only one of which contained an acoustically conveyed phrase boundary in the same place as in the text sentence (see *Figure 1* for schematic and example sentences). Trials were scored as correct if a participant pressed the button associated with the auditory sentence that correctly matched the text sentence. Proportions of correct judgments (*Figure 2*) were subjected to a repeated-measures analysis of variance. Overall, proportion correct across amusia and control groups was matched (main effect of Group, $F(1,84) = 0.16$, $p=0.69$, interaction of Group by Condition, $F(2,84) = 0.374$, $p=0.96$). This lack of interaction was predicted based on previous results obtained from a similar paradigm using out-of-scanner data but from the same participants (*Jasmin et al., 2020a*). There was a main effect of condition ($F(2,84) = 3.32$, $p=0.04$). Follow-up post-hoc testing indicated that performance in the Both-Informative condition (with pitch and duration cues simultaneously present) was more accurate than either Pitch-Informative ($t(84) = 2.31$, $p=0.023$) or Duration-Informative ($t(84) = 2.15$, $p=0.03$), a result that was also predicted and which

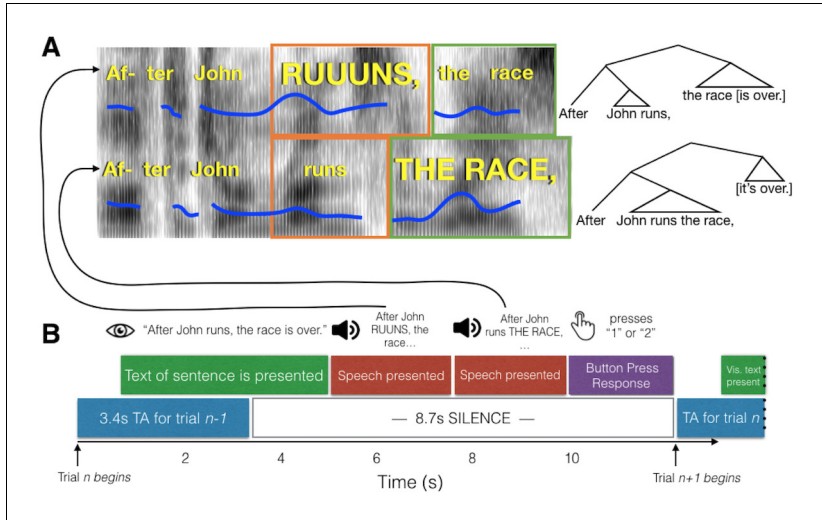

**Figure 1.** Schematic of experimental paradigm. (**A**) Example spectrograms of the early closure (top) and late closure (bottom) stimuli for the Both-Informative condition. Fundamental frequency contours are indicated with blue lines. The relative duration of the critical words are indicated with orange and green boxes. To the right, syntactic trees for the two sentences are shown to highlight the grammatical structure indicated by the phrase boundaries. (**B**) The time course of a single trial. Participants read a text version of the sentence from the screen, which was either early or late closure. This was followed by auditory presentation of the late and early closure versions. After both recordings were played, participants chose whether the first or second recording they heard matched the visually presentence sentence better. A single whole-brain volume was acquired after the button press, timed to capture the peak of the hemodynamic response roughly around presentation of the second sentence.

replicates the behavioral findings in *Jasmin et al., 2020a*. One outlier control participant's performance was less than 0.3. Re-analysis of the data without this participant did not change the results pattern.

## Neuroimaging - whole-brain connectedness

Results from these analyses are available online (see Data Availability Statement for details). A data-driven approach was taken to identify brain regions with the largest group- and condition-related differences in functional connectivity (see Materials and methods). Comparing whole-brain connect-edness values by group (Amusia vs. Controls) revealed four significant locations (where *z* of peak vertices > 4.61, FDR-corrected p<0.05) that showed greater whole-brain connectedness for the control than for the amusia group (see *Figure 3*, yellow crosses). All group differences were located in the inferior frontal cortex: two left hemisphere vertices (inferior frontal gyrus *p. triangularis* and dorsolateral prefrontal cortex) and two right hemisphere vertices (inferior frontal gyrus *p. triangularis* and *p. orbitalis*). There were no areas where whole-brain connectedness differed by Condition, or showed an interaction of Group and Condition.

## Follow-up seed-to-whole brain tests

Follow-up testing was conducted on the four significant regions (Control > Amusia, collapsed across the three conditions) identified above to characterize the specific cortical regions driving these group connectivity differences (*Berman et al., 2016*; *Gotts et al., 2012*; *Jasmin et al., 2019*; *Song et al., 2015*). Relative to control participants, amusic participants' left inferior frontal gyrus seed region showed particularly notable decreases in connectivity with the right posterior superior temporal and inferior parietal cortex, as well as with the right posterior superior temporal sulcus (*Figure 3A*). Analysis of subcortical connectivity indicated that there was also weaker connectivity with the right nucleus accumbens (*Table 1*).

The left dorsolateral prefrontal cortex in amusic participants showed decreased functional connectivity with the mid portions of the right superior temporal gyrus, posterior part of the right

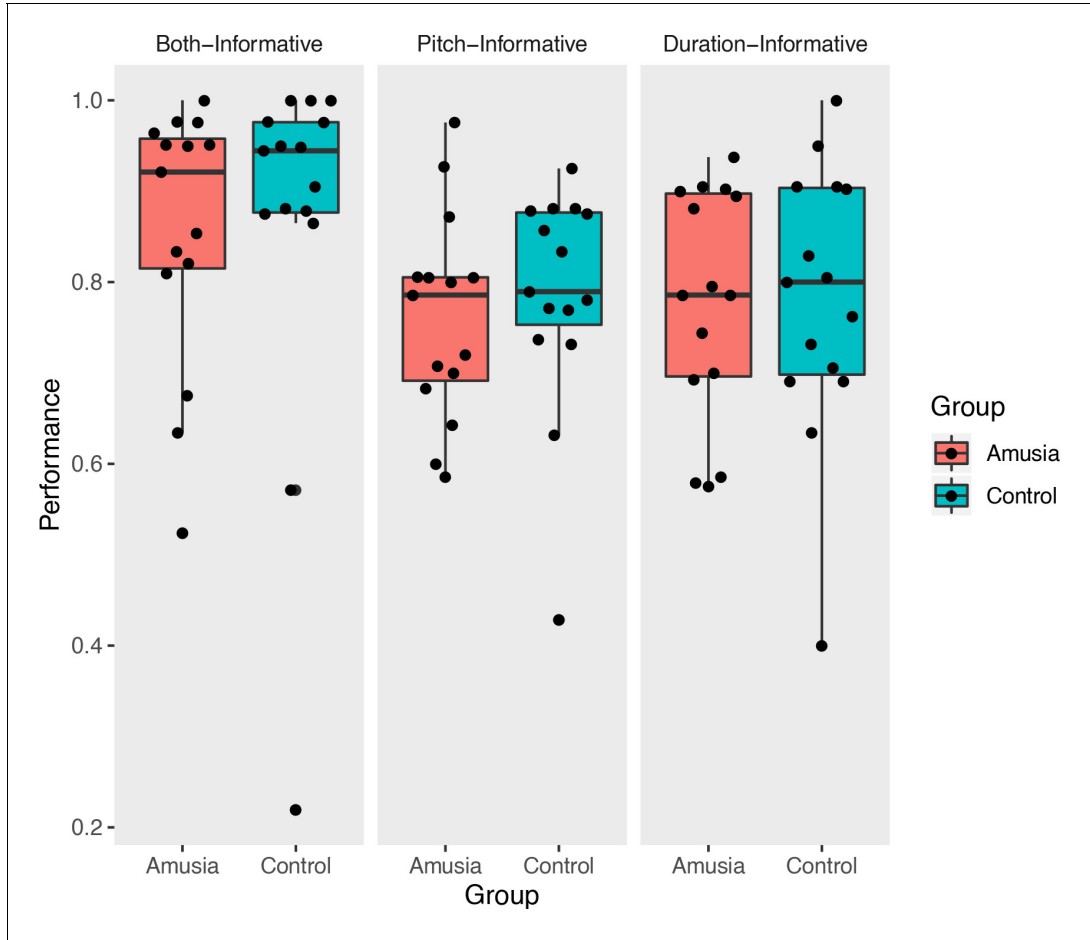

**Figure 2.** In-scanner performance. Prosodic categorization performance measured in the scanner (proportion correct); each point represents the performance of a single participant.

The online version of this article includes the following source data for figure 2:

**Source data 1.** n-scanner performance.

middle temporal gyrus extending into the inferior bank of the superior temporal sulcus, and the right anterior insula (*Figure 3A*). Several subcortical structures - bilateral caudate nucleus and putamen, bilateral pallidum, bilateral cerebellum, and bilateral thalamus - also showed significantly reduced (FDR-corrected) connectivity with the seed in amusics (*Table 1*).

The right *pars triangularis* seed showed Control > Amusic connectivity with right dorsolateral prefrontal cortex and left posterior superior temporal gyrus (*Figure 3B*). It also showed decreased connectivity with left nucleus accumbens. Right *pars orbitalis* showed decreased connectivity with right dorsolateral prefrontal cortex (*Figure 3B*). There was also decreased connectivity with the left thalamus (*Table 1*).

## Correlations between functional connectivity levels and prosodic cue weights

Of the 30 participants in this study, 21 took part in an experiment that measured the degree to which they relied on pitch versus duration to categorize prosody, that is, their 'normalized prosodic cue weights', which ranged from 0 to 1, with values greater than 0.5 indicating greater reliance on pitch than duration, and values less than 0.5 indicating greater reliance on duration than pitch (Experiment 1, *Jasmin et al., 2020a*). These cue weights were assessed with respect to the functional connectivity results reported above. Across this subset of participants, normalized cue weights were correlated with L-DLPFC <=> R insula connectivity (Spearman R = 0.78, p=0.000037), and

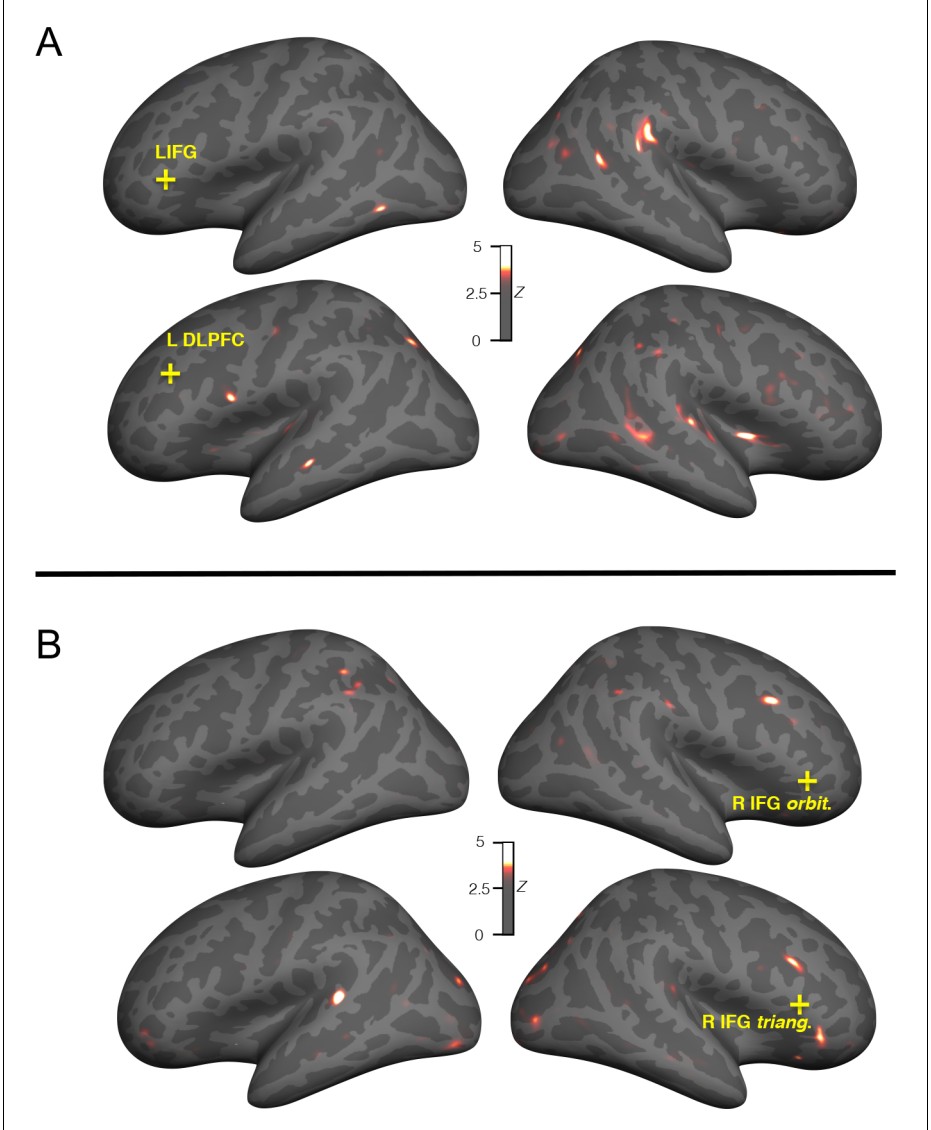

**Figure 3.** Seed locations and group differences in seed-to-whole brain functional connectivity. Inflated surfaces show the locations of False Discovery Rate-corrected group differences (Control > Amusia) in whole-brain connectivity (yellow crosses, minimum Z > 4.61), which were used as seeds in subsequent analyses (minimum Z > 3.57; warm colors indicate greater connectivity in the control than amusia participants). All four seed vertices were located in inferior frontal cortices (left inferior frontal gyrus, left DLPFC, right inferior frontal gyrus *p. triangularis*, and right inferior frontal gyrus *p. orbitalis*) (**A**) Significant group differences (Control > Amusia) in functional connectivity with left hemisphere seeds. The largest decreases in connectivity in the amusia group were located in right superior temporal plane and gyrus, the posterior middle temporal gyrus onto the inferior bank of the superior temporal sulcus, and anterior insula. (**B**) Significant group differences (control vs amusia) in functional connectivity with right hemisphere seeds. Prominent decreases in connectivity with the right inferior frontal gyrus in individuals with amusia were observed in the superior temporal plane and regions of occipital, frontal, and parietal cortex.

L-DLPFC <=> R auditory cortex connectivity (Spearman R = 0.75, p=0.000154; *Figure 4*). This indicated that participants who relied least on pitch information to process speech had the weakest functional connectivity between these areas, while those who relied most on pitch had the strongest.

Although analyzing the control and amusic groups independently results in extremely small sample sizes, this pattern also held (albeit with 'marginal significance') within the 11 control participants alone, for both auditory cortex connectivity (R = 0.58, p=0.06) and insular connectivity (R = 0.55,

**Table 1.** Significant main effects of Group involving functional connectivity between seed areas and subcortical Structures.
All effects are Control > Amusia.

| Seed | Region of interest | F(1,87) | p |
|---|---|---|---|
| L IFG | R Accumbens | 15.43 | 0.0002 |
| L DLPFC | L Putamen | 15.78 | 0.0001459 |
| | R Putamen | 17.78 | 0.00006047 |
| | L Caudate | 25.23 | 0.0000027 |
| | R Caudate | 11.51 | 0.001044 |
| | L Cerebellum | 24.47 | 0.00000364 |
| | R Cerebellum | 16.23 | 0.0001194 |
| | L Pallidum | 14.60 | 0.0002484 |
| | R Pallidum | 12.44 | 0.0006739 |
| | L Thalamus | 14.83 | 0.0002245 |
| | R Thalamus | 15.72 | 0.0001501 |
| R IFG (orbit) | L Thalamus | 14.83 | 0.0002245 |
| R IFG (triang.) | L Accumbens | 10.10 | 0.002054 |

p=0.08). Both these correlations were in the predicted direction, suggesting that even non-amusics may perform dimensional reweighting of acoustic dimensions and functional connectivity. Correlations within the (much more variable) amusic group alone were weaker and non-significant (although again, the group size is very small).

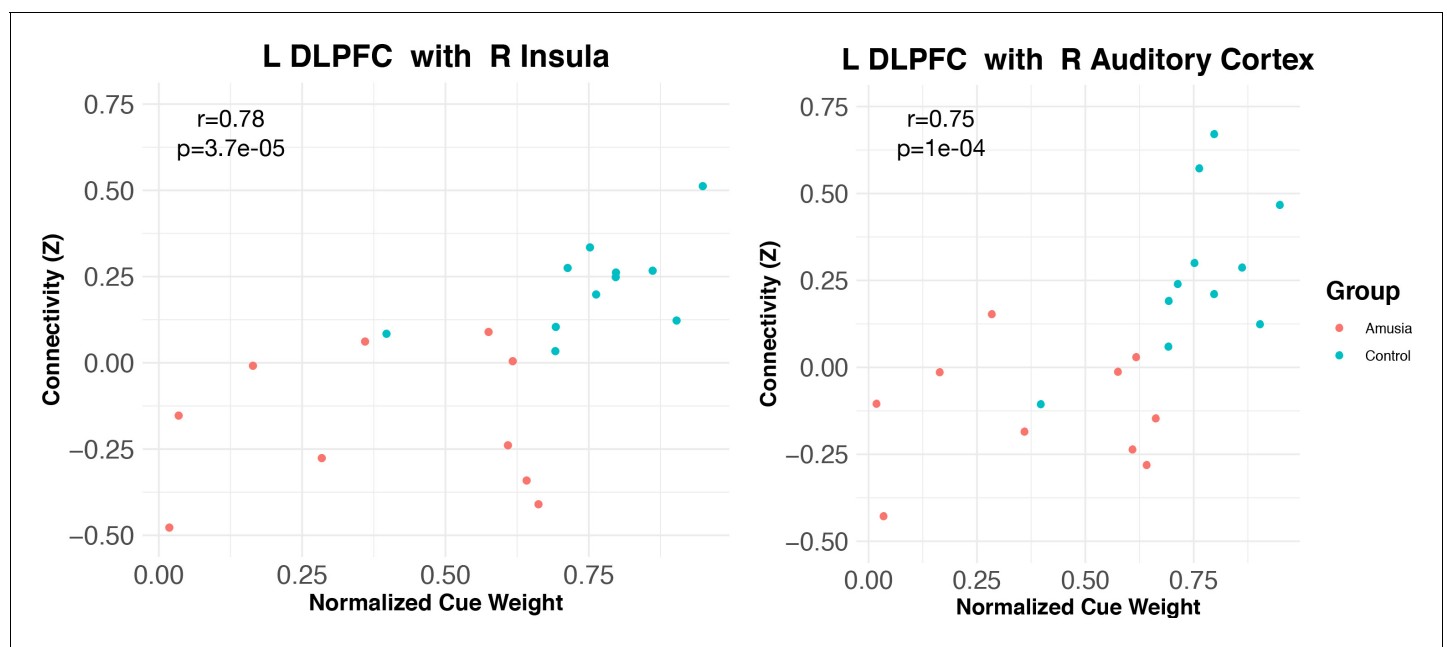

**Figure 4.** Connectivity between L DLPFC and insular (left) and auditory (right) cortex is modulated by normalized cue weights measured outside the scanner. Correlation coefficients are Spearman rho.
The online version of this article includes the following source data for figure 4:

**Source data 1.** Brain-behaviour correlations.

## Comparison with task-free data

To ensure that the pattern of connectivity we observed between groups (decreased right auditory cortex and right insula with L-DLPFC connectivity) was not due to intrinsic, task-irrelevant differences in neural architecture, the data from the language task was compared to that collected during passive listening to tone sequences. Whereas during speech perception amusic subjects showed reduced functional connectivity between left frontal and right insula/auditory ROIs relative to controls (p=0.0001 for both ROIs; in line with the whole-brain imaging analyses), this pattern did not hold during passive listening to tones (Amusia vs Control connectivity, p=0.29, Group (Amusic, Control) by Task (Speech Perception, Passive Tone Listening) interaction p=0.045 for the insula ROI; Amusia vs Control p=0.30, Group by Task interaction p=0.035 for the auditory cortex ROI - see *Figure 5*). These interactions suggest that our neural connectivity results are specifically linked to speech perception, rather than reflecting an overall connectivity difference between groups regardless of task state.

## Activation results for the speech processing task

Although we were concerned with functional connectivity rather than activation, we also tested for differences in activation levels between groups and conditions. False Discovery Rate correction was used to correct for multiple comparisons across both hemispheres for each test (Group, Condition and Group X Condition). No significant differences were detected for the main effects of group and condition, nor the interaction of those factors.

## Discussion

We found that individuals with amusia, who have been previously shown to rely less on pitch than controls to process spoken language (*Jasmin et al., 2020a*), exhibited decreased functional connectivity between left frontal areas and right hemisphere pitch-related regions. In our task, participants matched spoken sentences with visually presented sentences based on pitch, duration, or both these acoustic dimensions together. Using a data-driven approach, we identified four regions in left and

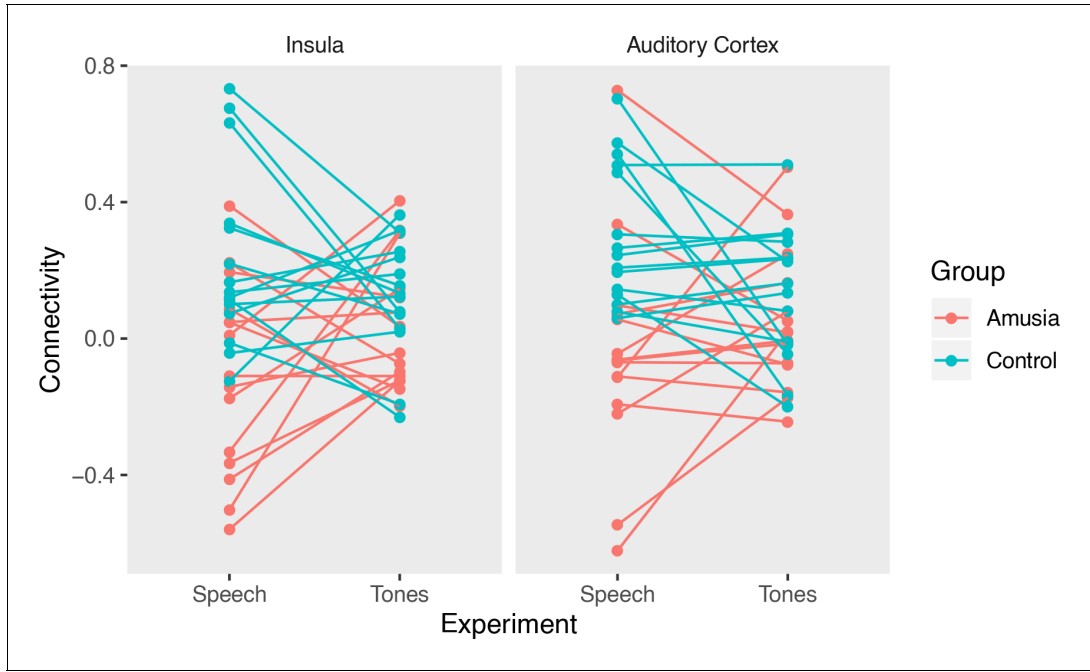

**Figure 5.** Connectivity between L DLPFC and right insula (left) and between L DLPFC and right auditory cortex (right) were reduced in the amusia group during speech perception (Control >Amusia, p=0.0001 for both ROI pairs), but not during passive tone perception.

The online version of this article includes the following source data for figure 5:

**Source data 1.** LDLPFC-ROI connectivity.

right inferior frontal cortex for which the amusic group exhibited decreased functional connectivity with several other sites in frontal, temporal and occipital cortex. The most prominent of these results was decreased connectivity between left frontal regions classically implicated in language processing (left IFG and DLPFC) and right hemisphere regions —in the superior temporal gyrus and sulcus, Heschl's gyrus, and anterior insula—that have been implicated in pitch processing (*Lee et al., 2011*; *Garcea et al., 2017*; *Warren et al., 2003*; *Hohmann et al., 2018*). We suggest that this decreased connectivity between right hemisphere pitch and left hemisphere frontal cortices may relate to the unreliability of the amusics' perception of and memory for pitch. This is similar to the 'weighted connections' model of multisensory integration, where a more (or less) reliable modality is given a stronger (or weaker) weight (*Beauchamp et al., 2010*).

Congenital amusia is often described as a disorder related to structural and functional connectivity within the right hemisphere, particularly between right inferior frontal cortices and right posterior temporal cortex (see *Peretz, 2016* for review). Consistent with this proposal, we found in the present study that right inferior frontal cortex exhibited strongly decreased functional connectivity in the amusia group, and follow-up seed testing revealed that right auditory areas were involved as well. However, we also found that sites in *left* frontal cortex also showed large decreases in connectivity in amusia, also most prominently with right hemisphere auditory areas. Our results are consistent with an account that right hemisphere auditory areas are not only abnormally connected to right frontal areas (as observed during tonal tasks) but are less integrated with frontal left hemisphere regions when processing speech and language.

Our null results for group differences in activation during speech processing are consistent with prior reports that amusics and controls do not differ in pitch representations within sensory regions. For example, the extent of pitch-responsive regions within auditory cortex has been shown to be similar in participants with amusia and controls (*Norman-Haignere et al., 2016*). Brainstem encoding of pitch in speech and musical stimuli is similarly unimpaired in individuals with amusia (*Liu et al., 2015b*). Moreover, in oddball EEG paradigms, amusics show similar pre-attentive mismatch negativity responses to small pitch deviants, but impaired attention-dependent P300 responses (*Moreau et al., 2009*; *Peretz et al., 2009*; *Mignault Goulet et al., 2012*; *Moreau et al., 2013*). These findings, along with the fact that amusics show intact non-volitional behavioral responses (unconscious pitch shifts) when presented with pitch-altered feedback of their own voice (*Hutchins and Peretz, 2012*), have been interpreted as evidence that amusia is a disorder of pitch awareness rather than one of low-level pitch processing (*Peretz et al., 2009*), with differences in structural connectivity as one possible foundation of this putative impaired pitch awareness (*Hyde et al., 2006*; *Loui et al., 2009*; but see *Chen et al., 2015*).

Our interpretation of differences in functional connectivity between amusics and controls diverges somewhat from these previous approaches: we argue that down-weighting of pitch information during perceptual categorization in both speech and music is adaptive, inasmuch as amusics have learned that pitch is an unreliable source of evidence relative to other perceptual dimensions. The evidence above suggesting that encoding of pitch in the brainstem and auditory cortex and pre-attentive responses to pitch changes are unaffected in amusia can be interpreted as suggesting that the fundamental deficit in amusia may not be increased perceptual noise or decreased pitch awareness but difficulties with retention of pitch information in memory (see *Tillmann et al., 2016* for review). Our task arguably taxed working memory resources: in a similar paradigm performed by the same participants in quiet listening conditions (*Jasmin et al., 2020a*), the mean reaction time measured from the end of the second auditory stimulus was 1.64 s, indicating that participants needed some time to compare both auditory presentations and make their judgments. This interpretation is consistent with evidence suggesting that amusics have difficulty with pitch sequence processing tasks even when discrimination thresholds are accounted for (*Tillmann et al., 2009*), as well as the finding that delaying the time interval between standard and comparison tones exacerbates pitch discrimination impairment in individuals with amusia (*Williamson et al., 2010*). Moreover, the pitch awareness account of amusia cannot explain the *Jasmin et al., 2020a* finding that pitch cues are downweighted only during longer-scale suprasegmental speech perception, while pitch weighting is not different between amusics and controls during shorter-scale segmental speech perception, despite pitch cues being arguably more subtle in the segmental condition. However, this finding can be explained by the pitch memory account, as the suprasegmental task requires detection of and memory for pitch patterns within a complex sequence, while the segmental task does not.

Furthermore, an account of amusia which suggests that the disorder primarily stems from differences in structural connectivity cannot account for the recent finding that functional connectivity patterns do not differ between amusics and controls during a verbal memory task (*Albouy et al., 2019*), as well as our finding that amusics and controls show similar functional connectivity patterns during passive listening to tone sequences. We suggest, therefore, that amusics neglect pitch because they have implicitly learned that their memory for pitch is unreliable, and that this down-weighting of pitch is reflected in decreased functional connectivity between right auditory areas and downstream task-relevant areas which integrate information from perceptual regions. One way to test this hypothesis would be to examine functional connectivity during perceptual categorization of consonant-vowel syllables as voiced versus unvoiced based on a pitch cue (F0 of the following vowel) and a durational cue (voice onset time). We predict, based on our previous findings (*Jasmin et al., 2020a*), that functional connectivity will not differ between amusics and controls on this task, a finding which would not be predicted by the pitch awareness account of amusia.

We note that a previous fMRI study on amusia detected group differences in functional connectivity during passive listening to tones. That study used task-defined seed voxels in bilateral auditory cortex and found, in the amusia group, increased connectivity between left and right auditory cortex, but decreased connectivity between right auditory cortex and right inferior frontal gyrus (*Hyde et al., 2011*). The present study does not necessarily clash with these findings, as we used different seed ROIs selected with a different procedure.

We did not observe any differences in functional connectivity between conditions in our speech task. This may be because our functional imaging protocol was timed to capture the peak in the BOLD signal corresponding to the presentation of the second auditory stimulus. Participants never knew (even implicitly) which acoustic dimension might be useful on any given trial until after they had heard both spoken sentences and needed to compare them to make their response. Furthermore, pitch fluctuations in the stimuli were above participants' thresholds , even in the Duration-Informative condition (where the standard deviation of F0 over each spoken utterance was, on average, 2.7 semitones), and so it is unsurprising that functional connectivity did not change on a trial-by-trial basis, and instead the same 'neural strategy' was employed to process speech regardless of the trial type.

Several other future directions are suggested by our results, particularly for examining cue weighting during auditory/speech perception. In the multimodal integration studies mentioned above (*Beauchamp et al., 2010*; *Nath and Beauchamp, 2011*), reliability of two different sensory modalities was manipulated experimentally by severely degrading input channels with noise, resulting in changes in connectivity. Similarly, aspects of speech could be selectively masked with noise in order to make them less reliable, which in turn could cause corresponding changes in functional or effective connectivity. Indeed, behavioral work has indicated that when fundamental frequency (pitch) or durational aspects of speech are manipulated to be unreliable cues, categorization behavior shifts such that participants place less relative weight on the dimension that has been made less reliable (*Holt and Lotto, 2010*). Certain groups, such as tone language speakers, are known to have fine-grained pitch perception abilities, and tend to place greater weight on pitch even when processing speech from a second, non-tonal language that they have learned (e.g. English; *Yu and Andruski, 2010*; *Zhang and Francis, 2010*, *Zhang et al., 2008*; *Qin et al., 2017*; *Jasmin et al., 2020a*). Given the increased reliability of their pitch perception, tone language speakers may exhibit correspondingly high connectivity strength between right hemisphere auditory regions and left hemisphere 'language regions' when pitch cues are present (more so than native non-tonal language speakers). Expert musicians also have extensive pitch-related experience and training and could also serve as a population to examine in future work.

## Materials and methods

### Participants

Participants, 15 individuals with amusia (10 F, age = 60.2 ± 9.4, range = 43–74) and 15 controls (10 F, age = 61.3 ± 10.4, range = 38–74), were recruited from the UK and were native British English speakers. The amusic group sample size reflected the maximum number of participants that could be screened and tested during our data collection period. The control group sample size was

matched to this. All participants gave informed consent, and ethical approval was obtained from the relevant UCL and Birkbeck ethics committees. Amusia status was obtained using the Montreal Battery for the Evaluation of Amusia (MBEA). Participants with a composite score (summing the Scale, Contour and Interval tests scores) of 65 or less were classified as having amusia (*Peretz et al., 2003*). We also note that the amusics defined using the MBEA had higher pitch thresholds than controls (Wilcoxon Rank Sum W = 29, p=0.001) but did not differ from controls in tone duration discrimination (W = 129, p=0.74), speech-in-noise threshold (W = 155.5, p=0.17), or audiometric hearing thresholds (t(28) = 1.33, p=0.20; see *Jasmin et al., 2020a* for detailed methods for these procedures).

## Stimuli

The stimuli were 42 compound sentences that consisted of a pre-posed subordinate clause followed by a main clause (see *Figure 1* for an example, and *Jasmin et al., 2020a*, *Jasmin et al., 2020b* for details). There were two versions of each sentence: (1) an 'early closure' version, where the verb of the subordinate clause was used intransitively and the following noun was the subject of a new clause ['After Jane dusts, the dining table [is clean]"]; and (2), 'late closure', where the verb was transitive and took the following noun as its object, moving the phrase boundary to a slightly later position in the sentence ['After Jane dusts the dining table, [it is clean]"]. The words in both versions of the sentence were identical from the start of the sentence until the end of the second noun ('After Jane dusts the dining table . . .''), and only the lexically identical portions of the sentences were presented to participants; thus the two stimuli did not differ in words spoken.

A native British English speaking male (who had previously trained as an actor) recorded early closure and late closure versions of each sentence in a sound-proofed room. The recordings were cropped such that only the portions with the same words remained, and silent pauses after phrase breaks were removed. Synthesized versions of these sentences were created with STRAIGHT voice-morphing software (*Kawahara and Irino, 2005*). First, the two versions of the sentence were manually time-aligned by marking corresponding 'anchor points' in the two recordings. Then, morphed speech was synthesized by varying the degree to which the early closure and late closure recordings contributed duration and pitch information. We synthesized pairs of stimuli in three conditions: (1) In the Pitch-Informative condition, the stimulus pair had exactly the same durational properties (that is, the length of phonemes, syllables, and words was the average between the two original recordings) but the vocal pitch indicated early or late closure at a morphing level of 80%; (2) in the Duration-Informative condition, vocal pitch in the stimulus pair was identical (at 50% between both versions) but the durational characteristics indicated early or late closure at a morphing level of 80%; (3) in the Both-Informative condition, both pitch and time cued early or late closure simultaneously at 80%. The morphed speech varied only in duration and pitch, while all other aspects of the acoustics (such as amplitude and spectral characteristics other than pitch) were the same, held constant at 50% between the two original recordings during morphing. This stimulus set is freely available (*Jasmin et al., 2020b*). Across all stimuli, F0 (vocal pitch) differences between early and late closure versions were large, with a mean of maximum difference of 7.7 semitones and range of 4.0–12.6 semitones. Thus, even the stimulus pair with the smallest pitch difference (4.0 semitones) exceeded the ~1.5 semitone pitch change detection threshold of the 'most impaired' participant in the amusia group (*Jasmin et al., 2020a*), which increased the chances that the amusia group would not suffer from poor performance, thereby avoiding a performance-related confound with our experimental design (see *Church et al., 2010* for discussion).

## MRI data collection

Subjects were scanned with a Siemens Avanto 1.5 Tesla magnetic resonance imaging scanner with a 32-channel head coil, with sounds presented via Sensimetrics S14 earbuds, padded around the ear with NoMoCo memory foam cushions. Functional data were collected using a slow event-related design with sparse temporal sampling to allow presentation of auditory stimuli in quiet. We used an echo planar image sequence, with 40 slices, slice time 85 ms, slab tilted to capture the entire cerebrum and dorsal cerebellum, ascending sequential acquisition; $3 \times 3 \times 3$ mm voxel size; silent stimulus and response period = 8.7 s, volume acquisition time = 3.4 s, total inter-trial interval = 12.1 s, flip angle = 90 degrees, bandwidth = 2298 Hz/pixel, echo time (TE) = 50 ms. After collecting functional

runs, a high-resolution T1-weighted structural scan was collected (MPRAGE, 176 slices, sagittal acquisition, 2x GRAPPA acceleration, 1 mm isotropic voxels, acquisition matrix = 224 × 256).

## Procedure (see schematic in *Figure 1*)

Each run began with three dummy scans to allow magnetic stabilization. Each trial (repetition time) lasted 12.1 s. The start of each trial was triggered by a pulse corresponding to the start of a volume acquisition (which acquired neural data from the previous trial, at a delay). At $t = 1$ s into the trial, the sentence appeared on the screen; before scanning participants were instructed to read each sentence silently to themselves. At $t = 5$ s (plus or minus a random 100 ms jitter) participants heard a spoken version of the first part of the sentence. At $t = 7.4$ s (plus or minus 100 ms jitter) the second version was presented. The two spoken versions contained the same words but their pitch and/or timing characteristics cued a phrase boundary that occurred earlier or later in the sentence. Following this, there were approximately 2 s of silence during which the participant responded with the button box, before the scanner began acquiring the next volume at $t = 12.1$ s. Participants performed three blocks of 42 trials (14 each of Pitch-Informative, Duration-Informative, and Both-Informative) with 8 Rest trials interspersed within each block.

## Comparison task - passive listening to tones

Following data collection for this task and the structural scan, participants took part in two task-free fMRI scanning runs in which they watched a silent film (*The General,* starring Buster Keaton, or an episode of the *Planet Earth* series played without sound) while being presented auditorily with semi-random tone sequences. Stimuli consisted of sequences of 'pips' - 30 ms 6-harmonic complex tones. The fundamental frequencies of the pips were either 440, 466.16, 493.88 or 523.25 Hz, and the time between tone onsets was 0.075, 0.125, 0.175, or 0.225 s. The transition probabilities (determining whether pip N+1 had the same pitch or duration properties as pitch N) were set at either 0.1 and 0.9 for duration and either 0.3 and 0.7 for pitch. These two transition parameters were 'crossed' to create four design cells, and 25 random sequences were generated for each cell. MRI scanning parameters were identical to those used in the active, prosody task, except the time between volume acquisitions was 17.1 s. Participants listened to 100 tone sequences across two runs (50 per run). Matlab code used to create the stimuli can be found online (see Data Availability Statement).

## MRI pre-processing

Image preprocessing was performed with FreeSurfer 6.0.0 (*Fischl, 2012*) and AFNI-SUMA 18.1.18 (*Cox, 1996*). Anatomical images were registered to the third echo planar image of the first run using Freesurfer's *bbregister* and processed with FreeSurfer's automated pipeline for segmenting tissue types, generating cortical surface models, and parcellating subcortical structures. Masks of inferior colliculi were obtained by manually examining individual subjects' anatomical images and selecting a single EPI voxel located at its centre, bilaterally. Freesurfer cortical surface models were imported to AFNI with the @SUMA_Make_Spec_FS program. Then a standard pre-processing pipeline using AFNI's *afniproc.py* program was used: all echo planar image volumes were aligned to the third repetition time of the first run using AFNI's 3DAllineate, intersected with the cortical surface with SUMA, smoothed along the surface with a 2D 6-mm-FWHM kernel, and converted to a standard mesh (std.141) for group analyses, separately for each hemisphere, where each vertex in the mesh (198812 per hemisphere) is aligned to the 'same' location in the cortex across subjects, using curvature-based morphing. Preprocessing of the passive listening experiment data was identical.

## Motion

The magnitude of transient head motion was calculated from the six motion parameters obtained during image realignment and aggregated as a single variable using AFNI's @1dDiffMag to calculate a Motion Index (*Berman et al., 2016*; *Gotts et al., 2012*; *Jasmin et al., 2019*). This measure is similar to average Frame Displacement over a scan (*Power et al., 2012*) and is in units of mm per repetition time. The difference in average motion between the groups was small (amusia group mean motion = 0.31 mm/TR; control group mean = 0.28 mm/TR) and amounted to 32 micrometers (~1/30[th] of a millimeter) per TR. The mean and distribution of motion did not differ statistically between groups (two sample *t*-test p=0.70, two-tailed).

## Beta series analysis of context-modulated functional connectivity

Given the previous reports (described above) of changes in connection strength between unimodal and multimodal areas in response to noise (*Beauchamp et al., 2010*; *Nath and Beauchamp, 2011*), we chose a connectivity-based analysis approach for our study. *Beta series correlation* (*Rissman et al., 2004*) is a technique for examining functional connectivity and its modulation by task, using correlations in trial-by-trial responses. It has been shown to be more powerful than alternatives such as generalized psycho-physiological interaction (gPPI) for event-related designs (*Cisler et al., 2014*). In a beta series analysis, one beta weight is calculated for each trial in the experiment (rather than for each condition). All the trial-wise betas associated with a given condition are then serially ordered to form a 'beta series'. Finally, using the beta series in the same way as a standard BOLD fMRI time series, functional connectivity (measured as Pearson correlations) is calculated between seed regions of interest and the rest of the brain. Differences in functional connectivity can then be examined by comparing groups, comparing conditions, or examining the interaction of these factors.

## Obtaining trial-wise beta weights

Our experiment used a slow event-related design with a long repetition time (12.1 s) and sparse temporal sampling (with volume acquisition separated by silent periods). Therefore, the time between acquisitions was long enough for the haemodynamic response to return to baseline, and each echo planar image acquisition corresponded to exactly one trial (*Figure 1*). For this reason, we did not convolve the echo planar image time series with a basis function during subject-level statistical analysis (*Hall et al., 1999*). In the design matrix for obtaining trial-wise betas, 126 column regressors were used (one for each non-rest trial). Each column vector was of length 150 (corresponding to all trials, including rest trials) and had a single 'one' in the position where the trial associated with that column occurred, while zeros were located in every other position. Polynomials up to second degree were also included in the model, on a run-wise basis, to remove the mean and any linear or quadratic trends. Fitting the trial regressors on a subject-wise basis resulted in cortical surface models of beta weights for each of the 126 trials, at each vertex on the reduced-vertex icosahedral cortical surface, with beta weights reflecting the neural response associated with that trial. As noted above, trial-wise betas were then serially ordered to form beta series separately for each of the three experimental conditions (Pitch-Informative, Duration-Informative, and Both-Informative) (*Rissman et al., 2004*). Because there were 30 participants, this procedure resulted in a total of 90 beta series (30 participants × 3 conditions=90 beta series). As for the passive tone listening data, because all 'trials' were of the same type, it was not necessary to separate them into conditions and perform a first-level model to obtain betas. However, polynomials up to second degree were detrended from the pre-processed data (as was done with the task data).

## Defining seed regions of interest

Beta series analysis requires initial seed voxels, vertices, or regions to be identified, whose trial-to-trial changes in activity are then compared to those of the rest of the brain. Rather than choose *a priori* seeds derived from the literature, which used mainly musical tasks or resting state, we used a data-driven approach to search for the largest group and condition differences in functional connectivity (*Berman et al., 2016*; *Cole et al., 2010*; *Gotts et al., 2012*; *Jasmin et al., 2019*; *Meoded et al., 2015*; *Song et al., 2015*; *Steel et al., 2016*; *Stoddard et al., 2016*; *Watsky et al., 2018*). To do this, we first calculated the 'whole-brain connectedness' of each cortical vertex (a procedure available in AFNI as the *3dTCorrMap* function). The whole-brain connectedness of a given vertex is defined as the Pearson correlation of activity within that vertex/voxel and the average signal across all neural gray matter in the rest of the brain. Mathematically, this is equivalent to calculating thousands of Pearson correlations, of a given vertex/voxel series and every other vertex/voxel series in the brain, and then taking the mean of those correlations (*Cole et al., 2010*), then repeating the process for every individual voxel/vertex. As such, it represents the global connectedness (or 'global correlation') of a vertex/voxel.

To calculate whole-brain connectedness, first, the average of trial-wise betas in gray matter across the brain was calculated in volume space, separately for each subject and for each condition, by running first-level (subject) models. The statistical models were identical to those conducted on the

cortical surface, described above, but were performed on volumetric Talairach images instead of the cortical surfaces. The reason for this choice was so that voxels in cortex and subcortex would contribute equally to our measure of global (whole-brain) connectivity. First, the average gray-matter beta value was calculated for each trial by intersecting each image in the beta series with a whole-brain gray matter mask (which excluded white matter and ventricles) and calculating the average beta value within the mask (*Gotts et al., 2012*; *Jasmin et al., 2019*). Next, this gray matter average was correlated with each cortical surface vertex's beta series, separately for each subject and condition, to obtain whole-brain connectedness maps. These values were then subjected to a statistical analysis based on our 2 (Group) × 3 (Condition) experimental design. Linear mixed effects models (AFNI's *3dLME*) (*Chen et al., 2013*) were constructed whose dependent variables were the vertex-wise whole-brain connectedness maps from each beta series. Group and Condition and their interaction were included as fixed effects. Participant was treated as a random intercept. Results of this step were corrected vertex-wise for multiple comparisons with False Discovery Rate ($q < 0.05$), separately for each test (Main Effect of Group; Main Effect of Condition; Interaction of Group by Condition) by pooling the p-values from both hemispheres' cortical surfaces. This False Discovery Rate threshold corresponded to uncorrected $p < 4 \times 10^{-6}$ for the Main Effect of Group. Four significant results (contiguous significant vertices) survived this threshold and were taken forward for the next analysis step. For the Main Effect of Condition and Interaction of Condition x Group, no results survived statistical correction at FDR ($q < 0.05$). An analogous procedure was run on the passive tone listening data, in which whole-brain connectedness values were compared by group (amusic vs. control) in a linear mixed effects model. No significant FDR-corrected group differences were detected, nor at a reasonable uncorrected threshold of $p < 0.001$.

A similar procedure was performed for subcortical structures. Beta series were obtained for each subject, structure, and experimental condition, from their standard Freesurfer subcortical parcellations by masking the EPI data within each structure and calculating the average of the voxels. Each structure's beta series was then correlated with the whole-brain gray matter beta average, separately for each condition, and the resulting values were subjected to linear mixed effects models with the same factors as above. Tests for Main Effect of Condition, of Group, and the Interaction of these factors was performed. All p-values were greater than $p > 0.001$ and no results survived an FDR-correction calculated over them.

## Follow-up seed-to-whole-brain testing

The first analysis step (seed definition, described above) identified which, if any, brain areas showed the largest connectivity differences between groups. However, this step is insufficient to localize the other specific regions driving this pattern. An analogy is in Analysis of Variance, where a significant omnibus test indicates a difference exists, but follow-up testing is required to determine where in the model differences exist (*Gotts et al., 2012*). Thus, to locate the regions driving this pattern, we undertook a second step: follow-up seed-to-whole-brain testing (*Cole et al., 2010*; *Gotts et al., 2012*; *Jasmin et al., 2019*). Each seed region was examined with respect to its connectivity pattern with every cortical vertex and subcortical structure.

For each of the 90 beta series (30 subjects by three conditions), values within the seed vertices were averaged and then correlated with the beta series for every vertex in the brain. These correlations were Fisher Z-transformed and used as the dependent variables in linear mixed effects models (*3dLME*) with the same fixed and random effects as above. For each of the seeds, we tested for the group difference (Amusia vs Control) in connectivity. Results were False Discovery Rate corrected to ($q < 0.05$) across all eight follow-up tests [four seeds × 2 hemispheres] corresponding to a threshold of $p < 0.00035$. Similarly, for the subcortical structures, each seed beta series was correlated with subcortical structure beta series, with resulting values subject to statistical testing. An FDR correction over all tests involving subcortex was applied. For display in figures, the data were converted from SUMA's standard mesh (std.141) to Freesurfer's standard surface (fsaverage) using AFNI's SurfToSurf program and mapping values from the closest nodes (i.e. vertices).

## Correlation between functional connectivity and cue weights

To determine whether the functional connectivity patterns we observed were related to the importance placed on acoustic dimensions during prosodic categorization (cue weighting), the functional

connectivity results were analyzed with respect to previously acquired cue weights obtained behaviorally from a subset of participants (*Jasmin et al., 2020a*). The right anterior insula and right auditory cortex results were used as ROIs (*Figure 3A*). The beta series for each ROI (averaged across vertices) was correlated with the beta series within the L-DLPFC seed area, separately for each condition, then averaged and Fisher Z-transformed. For the 21 participants for whom we had prosodic cue weight data (from *Jasmin et al., 2020a*), these cue weights were analyzed with respect to the functional connectivity between the L-DLPFC seed and the two ROIs using Spearman correlations.

## Comparison between the speech task and passive tone listening

As described above, functional connectivity between L-DLPFC, and right auditory cortex and right insula was calculated using data from the passive tone listening task, using ROIs derived from the active speech perception task. After pre-processing and de-trending, the averaged value from the tone listening experiment within these ROIs was extracted, as well as the LDLPFC seed, for each experiment. Correlations between signal within the seed and the two ROIs was calculated and Fisher Z-transformed. As mentioned above, because all trials in the tone-listening experiment were analyzed as the same type, it was not necessary to use a first-level model to obtain trial-wise betas. Similarly for the data from the speech task, the average value within the seed region and both ROIs was extracted, separately for each of the 3 Beta series (Pitch-, Time- and Both-Informative), and the seed and ROI series were correlated. The mean of these three correlation coefficients was calculated and Fisher Z-transformed. Finally, statistics were performed using a mixed ANOVA with Experiment (Speech or Tones) as the within-subject factor and Group (Amusia or Control) as the between-subject factor.

## Analysis of activation

A standard General Linear Model comparing activation strength (rather than connectivity) was also conducted. As in the General Linear Model for obtaining beta weights, no basis function was used, and polynomials up to second degree were included in the models.

## Data availability

The data that support the findings of this study are openly available in the Birkbeck repository (https://researchdata.bbk.ac.uk/65/), as are the speech stimuli (*Jasmin et al., 2020b*; https://researchdata.bbk.ac.uk/37/). The speech task can be demoed at the following link: (Gorilla Open Materials; https://gorilla.sc/openmaterials/102786).

# Acknowledgements

We thank our study participants.

# Additional information

### Funding

| Funder | Grant reference number | Author |
| --- | --- | --- |
| Wellcome | 109719/15/Z | Adam Taylor Tierney |
| Leverhulme Trust | ECF-2017-151 | Kyle Jasmin |
| Society for Education, Music and Psychology Research | | Kyle Jasmin |

The funders had no role in study design, data collection and interpretation, or the decision to submit the work for publication.

### Author contributions

Kyle Jasmin, Conceptualization, Data curation, Formal analysis, Investigation, Methodology; Frederic Dick, Conceptualization, Supervision, Visualization, Methodology; Lauren Stewart, Conceptualization, Resources; Adam Taylor Tierney, Conceptualization, Supervision, Funding acquisition

## Author ORCIDs

Kyle Jasmin https://orcid.org/0000-0001-9723-8207
Frederic Dick https://orcid.org/0000-0002-2933-3912
Adam Taylor Tierney https://orcid.org/0000-0002-7624-6918

## Ethics

Human subjects: All participants gave informed consent and ethical approval was obtained from the UCL Research Ethics Committee (fMRI/2016/001) and the Birkbeck Department of Psychology Research Ethics Committee (161711).

## Decision letter and Author response

Decision letter https://doi.org/10.7554/eLife.53539.sa1
Author response https://doi.org/10.7554/eLife.53539.sa2

---

# Additional files

## Supplementary files

• Transparent reporting form

## Data availability

The data that support the findings of this study are openly available in the Birkbeck repository (https://researchdata.bbk.ac.uk/65/), as are the speech stimuli (Jasmin et al., 2020b; https://research-data.bbk.ac.uk/37/). The speech task can be demoed at the following link: (Gorilla Open Materials; https://gorilla.sc/openmaterials/102786).

The following dataset was generated:

| Author(s) | Year | Dataset title | Dataset URL | Database and Identifier |
|---|---|---|---|---|
| Jasmin K, Dick F, Stewart L, Tierney A | 2020 | Altered functional connectivity during speech perception in congenital amusia | https://doi.org/10.18743/DATA.00065 | Birkbeck Research Data, 10.18743/DATA.00065 |

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
