## [Decision Letter]

**Acceptance summary:**

We found this paper to be an interesting demonstration of a link between behavior and functional brain connectivity when comparing people with amusia and a control group. The results suggest a potential neural basis for the down-weighting of pitch cues in people with amusia that has parallels with recent findings involving cue weighting between sensory modalities. The work opens up the possibility of further such studies that might tackle neural correlates of individual differences in cue weighting within a single sensory modality in the wider population.

**Decision letter after peer review:**

[Editors’ note: the authors submitted an appeal for reconsideration following the decision after peer review. What follows is the decision letter after the first round of review.]

Thank you for submitting your work entitled "Altered functional connectivity during speech perception in congenital amusia" for consideration by *eLife*. Your article has been reviewed by a Senior Editor, a Reviewing Editor, and two reviewers. The reviewers have opted to remain anonymous.

Our decision has been reached after consultation between the reviewers. Based on these discussions and the individual reviews below, we regret to inform you that your work will not be considered further for publication in *eLife*.

Although both reviewers found the scientific question and hypothesis very interesting and the data intriguing, both noted a lack of a brain-behavior relationship in the current study. It is possible that additional data could address the concerns outlined below. However, because of the substantial nature of such data collection, it would warrant a new submission rather than a revision.

Reviewer #1:

Jasmin et al. present an fMRI study on speech in congenital amusia. 15 amusics and 15 controls were scanned in the fMRI while they matched auditory and visual sentences, presumably using different acoustic cues.

I think the article sets out to address a fascinating puzzle. The puzzle is that although pitch information is in both music and language, people with amusia, who have trouble with pitch information in music, do not report difficulties in naturalistic speech perception. Here, the authors hypothesize that amusics calibrate their perception by down-weighing the cues that are less reliable for them, i.e. pitch, relying instead on other cues that co-occur with pitch.

I think the hypothesis is a fine and thoughtful one.

During task fMRI, amusics showed lower functional connectivity in left and right prefrontal regions, and between left prefrontal regions and right auditory and anterior insula. But the results as presented right now don't show any relationship between task performance and the neural findings. I would really like to see some brain-behavioral correlations to identify the specific compensatory mechanisms that amusics might be using to approach the task of sentence processing. WIthout that, it is unclear whether and how the findings in fMRI connectivity differences pertain to the task. Especially since there were no task-related differences in behavior nor were there any differences in task-related activations, one wonders if a task-free resting state fMRI would have shown the same results, thus suggesting intrinsic connectivity differences rather than addressing the differential weighting mechanisms so interestingly described in the Introduction.

Reviewer #2:

In this paper Jasmin et al. present fMRI data from 15 persons with amusia and 15 controls as they perform a linguistic task that requires judgments based either on pitch cues or duration cues, or both. The findings hinge on the fact that functional connectivity differs for the two groups, first in terms of global connectedness, in four frontal regions, two in each hemisphere, and second in terms of inter-regional functional connectivity between these frontal regions and posterior regions, including auditory areas. The authors interpret these findings in terms of their idea that persons with amusia down-regulate pitch cues because they know them to be unreliable, and this is reflected by the functional connectivity differences.

The study is well-conducted and analyzed, and the paper is well-written. But I am not persuaded about the interpretation of the results. The main problem is the lack of any explicit brain-behavior relationship. This issue takes two forms. First, the brain data and the behavioral data are never related to one another. The authors make the claim that the functional connectivity differences are related to the behavior, but this is not shown by any analysis. Without any such link, it is hard to know whether the connectivity differences observed have anything to do with the particular ability sampled by this task, or if they are related to something else entirely. Second, the behavior itself does not differ across participant groups. The authors specifically desired this outcome so that any imaging differences would not be driven by group behavioral differences. But that idea does not make sense to me because there were two conditions, one using pitch cues and another using duration cues. Presumably, amusic persons have trouble with the former but not the latter. Yet this was not the case in the behavior (no condition X group interaction), nor in the imaging data (the FC effects were not influenced by condition either). So if the FC results are somehow related to the downweighting of pitch cues it should show up specifically in the condition where such cues were available, and not in the condition where duration cues were available. Without this crucial dissociation, the interpretation does not hold together.

In the Discussion section the authors argue that the deficit in amusia has more to do with tonal retention than perception, an idea which is supported by other authors, although the bulk of the evidence suggests that both features are present. But it is not entirely clear why a retention deficit would be so important in the task used, which does not appear to require a lot of working memory since the spoken stimuli simply have to be matched to a visual representation of the sentence. Another related issue is that in amusia there is often a wide range of perceptual performance, with some individuals scoring near chance on pitch discrimination and others scoring at the same level as non-amusic persons. It's not clear how that heterogeneity may have influenced performance on this task, especially as the pitch intervals used in the stimuli are not documented in the manuscript, which seems like a significant omission. It would be important to know whether the excursions in F0 for the pitch-varying sentences (which are typically much larger than one or two semitones) were smaller than the thresholds that the amusics can perceive or not. In general, there is no discussion of individual differences in the data, which might have helped to understand the findings.

[Editors’ note: what follows is the editors’ decision letter after consideration of the letter of appeal. Further revisions were suggested prior to acceptance, as described below.]

Thank you for choosing to send your work entitled "Altered functional connectivity during speech perception in congenital amusia" for consideration at eLife. Your letter of appeal has been considered by a Senior Editor and a Reviewing editor, and we are prepared to consider a revised submission with no guarantees of acceptance.

Beyond the new data, please be sure to address other concerns of the reviewers, including the lack of interaction between group and condition in the behavior and in the imaging data. Perhaps the lack of interaction in the behavior is due to ceiling performance in both groups (please show the data), but if the functional connectivity results are related to the downweighting of pitch cues in amusics, shouldn't this only manifest itself in the condition where such cues were available, and not in the condition where duration cues were available? Ensuring that the data are fully compatible with the interpretation will play a critical role in a final decision.

[Editors’ note: further revisions were suggested prior to acceptance, as described below.]

Thank you for submitting your revised article "Altered functional connectivity during speech perception in congenital amusia" for consideration by eLife. Your article has been reviewed by two peer reviewers, and the evaluation has been overseen by Andrew Oxenham as Reviewing Editor and Barbara Shinn-Cunningham as the Senior Editor. The following individuals involved in review of your submission have agreed to reveal their identity: Anne Caclin (Reviewer #3).

The reviewers have discussed the reviews with one another and the Reviewing Editor has drafted this decision to help you prepare a revised submission.

We would like to draw your attention to changes in our revision policy that we have made in response to COVID-19 (https://elifesciences.org/articles/57162). Based on the reviews and the discussions between myself and the reviewers, we have come to the conclusion that the manuscript is potentially publishable in *eLife* but will need some further revision before it can be accepted. The areas where revisions are necessary are listed below.

Essential revisions:

Brain-behavior correlations. The inclusion of the current Figure 5 certainly strengthens the manuscript and is an important addition. However, because the correlation is driven primarily by the between-group differences, the importance of the overall correlation should not be overstated. In particular, it is probably not appropriate to mention it in the abstract without the necessary caveats. Instead just a statement, saying that the between-group differences in connectivity reflect the between-group differences in performance in the same groups of listeners, would suffice.

Behavior task. There are still some questions remaining on this front. In particular, there remain some questions as to why the amusics should show the same amount of behavioral gain when pitch cues are added. Is this thought to be because the pitch changes are so large as to influence performance despite impaired pitch processing? A little more discussion of this aspect of the results will likely help readers follow the logic better. There is also some skepticism regarding when and whether participants knew which cues were informative on each trial (Discussion section). For instance, in many cases participants would probably have been able to complete the task following just the first interval, rather than have to wait for the second interval and make a comparison. Although the 2AFC paradigm makes the analysis of performance easier, it cannot be assumed that participants always make full use (or need to make use) of the information from both intervals. This observation has some implications for the extent to which memory processes are involved.

Additional analysis. There are two areas where additional analyses may be illuminated. First, the main conclusions of the study rely in part on the results of the correlations between pitch cue weights and connectivity measures. Duration cue weights, which do not differ between groups, were also measured. It appears important, as a negative control, to comment on the absence (or not) of correlations between these duration cue weights and connectivity measures (subsection “Correlations between functional connectivity levels and prosodic cue weights”).

Second, the comparison of connectivity between the active and passive (tones) task was based on the ROIs derived from the passive task, with the assumption that the passive task should not result in differences in connectivity. However, as far as we could tell, the initial data-driven connectivity tests were not run separately on the passive task. If that were done just on the passive task, would any connectivity differences between the groups emerge? If not, that would provide a further important negative control.

---

## [Author Response]

[Editors’ note: the authors appealed the original decision. What follows is the authors’ response to the first round of review.]

Although both reviewers found the scientific question and hypothesis very interesting and the data intriguing, both noted a lack of a brain-behavior relationship in the current study. It is possible that additional data could address the concerns outlined below. However, because of the substantial nature of such data collection, it would warrant a new submission rather than a revision.

Both you and the reviewers flagged up the lack of a reported brain-behaviour correlation as the primary weakness of the study, and you suggested that because collecting the data required would be a substantial task and revision, a new submission would be warranted. Reviewer 1 also noted a lack of comparison with another task or with resting state as a weakness.

Because the reviewers seemed otherwise quite enthusiastic about the work, we thought we should write to make you aware that we have existing data that speaks directly to both of these issues, collected for other studies, but with the same participants. In retrospect, we should have included these analyses originally. The only reason we did not was in the interest of scope. Nevertheless, we agree with both reviewers that these are very pertinent to our interpretation and are pleased to include them now.

1) Brain-behaviour correlations

Of the 30 participants in the study we submitted to *eLife*, 21 took part in an experiment that measured the degree to which they relied on pitch versus duration to categorize prosody, i.e. their ‘normalized prosodic cue weights’ (Experiment 1, Jasmin, et al., 2019, Journal of Experimental Psychology: General). We have analyzed the key results from the imaging analysis in the present work – connectivity between left dorsolateral prefrontal (L-DLPFC) and right auditory and insular cortices during speech perception – with respect to these cue weights, within these 21 participants. Across this subset of participants, normalized pitch cue weights were associated with L-DLPFC-R auditory cortex connectivity (Spearman R = 0.78, p=.000037), and LDLPFC-R insula connectivity (Spearman R = 0.75, p=.000154; Figure 5). We used non-parametric analyses because the data appear to be heteroskedastic, but the Pearson correlations are also significant.

Although analyzing the control and amusic groups independently results in extremely small sample sizes, this pattern also held (albeit with “marginal” significance) within the 11 control participants alone, for both auditory connectivity (R = 0.58, p=.06) and insular connectivity (R=.55, p=.08). These were both in the predicted direction, suggesting that even non-amusics may perform dimensional reweighting. Correlations within the (much more variable) amusic group alone were weaker and non-significant (although again, the group size is very small).

Since the phenomenon we set out to explain in this study is the downweighting of pitch cues in amusia, it seems natural that cue weights themselves are the most appropriate behavioural measure to compare with the neural data. They were also measured with a high degree of precision, in a quiet laboratory setting and over 490 individual trials. However, we note that reviewer 2 suggested that we correlate our in-scanner behavioural measure with the neural data. The reason we did not use this data is that the in-scanner test was designed to be relatively easy, with the intention of matching the groups on performance, so as to avoid the confound of in-scanner behavioural differences potentially driving differential activation (a topic often discussed in the developmental fMRI literature, e.g., Church, Schlaggar and Petersen, 2010). Indeed, the groups did not differ in performance, even on the Pitch condition.

As a further ‘check’ that the neural differences were related specifically to cue weighting, we also correlated these ROI data with data from a test of the ability to perceive linguistic focus from pitch cues, a measure on which these participants had previously been shown to differ (Jasmin et al., 2019). These correlations were not significant across all 30 participants, either for the insula (r = 0.2, p=0.3) or auditory cortex data (r = 0.14, p=0.46).

2) Comparison with task-free data

Reviewer 1 flagged that our results could have reflected an ‘intrinsic’ difference in connectivity that might persist even in a resting state, in which case one could not claim that they are specifically related to more active speech or auditory processing perception. We have (unpublished) additional data that speaks to this issue. The 30 participants in our study also took part in an experiment in which they passively listened to tone sequences while watching a silent film during a ‘sparse scanning’ protocol similar to that used during the active task we report in the paper, and with an identical EPI sequence. We have taken the L-DLPFC seed and the auditory cortex and insula ROIs, extracted the average time series within them, and compared these time series to corresponding data from the speech perception task. Whereas during speech perception, amusic subjects showed much less functional connectivity between left frontal and right insula/auditory ROIs relative to controls (p=.0001 for both ROIs; in line with the whole-brain imaging analyses), this pattern did not hold during passive listening to tones (Group by Task interaction p = 0.045 for the insula ROI, p = 0.035 for the auditory cortex ROI – see Figure 6). These interactions suggest that our neural connectivity results are specifically linked to speech perception, rather than reflecting an overall connectivity difference between groups regardless of task.

[Editors’ note: what follows is the authors’ response to the additional feedback provided by the editor, after the authors’ appeal.]

Beyond the new data, please be sure to address other concerns of the reviewers, including the lack of interaction between group and condition in the behavior and in the imaging data.

Below we have addressed the concerns about the lack of interactions, as well as the secondary issues.

Perhaps the lack of interaction in the behavior is due to ceiling performance in both groups (please show the data).

Performance on the task is now plotted in new Figure 1. This shows performance was not at ceiling for either the amusia or control group during the Pitch-Informative condition, so we suggest ceiling effects were not the cause of the lack of interaction. (Indeed, neither was there a ceiling effect in the version of this experiment we ran outside the scanner in quiet listening conditions (Jasmin et al., 2020)).

But if the functional connectivity results are related to the downweighting of pitch cues in amusics, shouldn't this only manifest itself in the condition where such cues were available, and not in the condition where duration cues were available?

This is an important point and we apologize for not being clearer. Pitch cues were always *available* on every trial, but pitch contour only conveyed information about sentence phrase boundaries in two of three conditions. In the manuscript, we have now renamed the conditions to make this clearer: Pitch-Informative, Duration-Informative, Both-Informative.

The lack of main effect of Condition, or Condition by Group interaction in our fMRI data is what one would predict given the specific task structure. Here, participants never knew – even implicitly – which acoustic dimension might be useful on a trial until after they had heard both spoken sentences and then compared them to make their decision. (Note that the fMRI protocol was timed to capture the BOLD signal peak related to the second auditory stimulus.) Thus, there was no overt acoustic cue that signalled whether duration or pitch was the diagnostic auditory dimension on each trial. We have added a paragraph to the Discussion section to clarify this:

“We did not observe any differences in functional connectivity differences between conditions. This may be because our functional imaging protocol was timed to capture the peak in the BOLD signal corresponding to the presentation of the second auditory stimulus. Participants never knew (even implicitly) which acoustic dimension might be useful on any given trial until after they had heard both spoken sentences and needed to compare them to make their response. Furthermore, pitch fluctuations in the stimuli were above participants’ pitch thresholds, even in the Duration-Informative condition (where the standard deviation of F0 over each spoken utterance was, on average, 2.7 semitones), and so it is unsurprising that functional connectivity did not change on a trial-by-trial basis, and instead the same ‘neural strategy’ was employed to process speech regardless of the trial type.”

Our interpretation of the group differences in functional connectivity is that they reflect a difference in task strategy between the amusic and control participants: amusics down-weight pitch relative to other dimensions during prosody perception (see Jasmin et al., 2020). When pitch and duration present conflicting information, one would predict that amusics will base their decision more on duration (relative to controls). However, if pitch cues are large enough, and duration cues are not informative, amusics should still succeed in the task (see Peretz et al., 2005) – while still weighting pitch information differently because that dimension is typically unreliable for them. To make sure our fMRI design was not confounded with performance differences (see Church et al., 2010 for discussion), pitch differences between stimulus pairs were designed to exceed all participants’ pitch discrimination thresholds, ensuring that the task was relatively easy for all participants, even for amusics on the Pitch-Informative condition. We have now made this explicit in the manuscript in subsection “Stimuli”:

“Across all stimuli, F0 (vocal pitch) differences between Early and Late closure versions were large, with a mean of maximum difference of 7.7 semitones and range of 4.0-12.6 semitones. Thus, even the stimulus pair with the smallest pitch difference (4.0 semitones) exceeded the ~1.5 semitone pitch change detection threshold of the ‘most impaired’ participant in the amusia group (Jasmin et al., 2020), which increased the chances that the amusia group would not suffer from poor performance, thereby avoiding a performance-related confound with our experimental design (see Church et al., 2010 for discussion).”

Importantly, inducing a shift in pre-existing perceptual cue weights in the laboratory appears to require extreme changes in co-occurrence statistics, for instance by completely reversing the correlation between cues (Idemaru and Holt, 2011), or severely degrading one of the input channels (Beauchamp et al., 2010; Nath and Beauchamp, 2011). Our manipulation did not resemble either of these processes – both pitch and duration changes were present across all trials, with only their relative task utility differing with condition. Given previous experimental results, this manipulation was likely too subtle to affect cue weights and functional connectivity that had developed from a lifetime of learning. However, it would be interesting in future work to manipulate stimulus statistics in order to assess the neural correlates of short-term changes in cue weighting (as we suggest in the Discussion section).

“Several other future directions are suggested by our results, particularly for examining cue weighting during auditory/speech perception. In the multimodal integration studies mentioned above (Beauchamp et al., 2010; Nath and Beauchamp, 2011), reliability of two different sensory modalities was manipulated experimentally, by severely degrading input channels with noise, resulting in changes in connectivity. Similarly, aspects of speech could be selectively masked with noise in order to make them less reliable, which in turn could cause corresponding changes in functional or effective connectivity.”

Ensuring that the data are fully compatible with the interpretation will play a critical role in a final decision.

We appreciate that both reviewers and editors raised this important issue and hope that our interpretation of the data has now been presented more clearly.

Reviewer #1:[…]During task fMRI, amusics showed lower functional connectivity in left and right prefrontal regions, and between left prefrontal regions and right auditory and anterior insula. But the results as presented right now don't show any relationship between task performance and the neural findings. I would really like to see some brain-behavioral correlations to identify the specific compensatory mechanisms that amusics might be using to approach the task of sentence processing. Without that, it is unclear whether and how the findings in fMRI connectivity differences pertain to the task.

Thank you for highlighting this point. Per our previous letter, we now include correlations between functional connectivity levels and cue weights measured outside the scanner. These correlations support our interpretation that individual differences in functional connectivity reflect idiosyncratic perceptual weighting strategies. The following is now included in the manuscript:

From the Materials and methods section:

“To determine whether the functional connectivity patterns we observed were related to the importance placed on acoustic dimensions during prosodic categorization (cue weighting), the functional connectivity results were analyzed with respect to previously acquired cue weights obtained behaviorally from a subset of participants (Jasmin et al., 2020a) The right anterior insula and right auditory cortex results were used as ROIs (Figure 3). The beta series for each ROI (averaged across vertices) was correlated with the beta series within the L-DLPFC seed area, separately for each condition, then averaged and Fisher Z-transformed. For the 21 participants for whom we had prosodic cue weight data (from Jasmin et al., 2020a) these cue weights were analyzed with respect to the functional connectivity between the L-DLPFC seed and the two ROIs using Spearman correlations.”

From the Results section:

“Of the 30 participants in this study, 21 took part in an experiment that measured the degree to which they relied on pitch versus duration to categorize prosody, i.e. their ‘normalized prosodic cue weights’ (Experiment 1, Jasmin, et al., 2020a). […] Both these correlations were in the predicted direction, suggesting that even non-amusics may perform dimensional reweighting of acoustic dimensions and functional connectivity. Correlations within the (much more variable) amusic group alone were weaker and non-significant (although again, the group size is very small).”

2) Especially since there were no task-related differences in behavior nor were there any differences in task-related activations, one wonders if a task-free resting state fMRI would have shown the same results, thus suggesting intrinsic connectivity differences rather than addressing the differential weighting mechanisms so interestingly described in the Introduction.

Thank you for the suggestion; we agree it’s important to show that these group differences in functional connectivity are task-specific, rather than intrinsic. We now include data from a task-free passive tone listening session in the same participants, and compare them statistically with the task-related functional connectivity measures initially reported. These analyses show that the group differences in connectivity elicited in the prosody perception task are not present in the passive tone listening task, suggesting that they reflect differential perceptual weighting during auditory language comprehension, rather than intrinsic connectivity differences:

From the Materials and methods section:

“[…] functional connectivity between L-DLPFC, and right auditory cortex and right insula was calculated using data from the passive tone listening task. After pre-processing and de-trending, the averaged value from the tone listening experiment within these ROIs was extracted, as well as the LDLPFC seed, for each experiment. Correlations between signal within the seed and the two ROIs was calculated and Fisher Z-transformed. As mentioned above, because all trials in the tone-listening experiment were analyzed as the same type, it was not necessary to use a first-level model to obtain trial-wise betas. Similarly for the data from the speech task, the average value within the seed region and both ROIs was extracted, separately for each of the 3 Β series (Pitch-, Time- and Both-Informative), and the seed and ROI series were correlated. The mean of these 3 correlation coefficients was calculated and Fisher Z-transformed. Finally, statistics were performed using a mixed ANOVA with Experiment (Speech or Tones) as the within-subject factor and Group (Amusia or Control) as the between-subject factor.”

From the Results section:

“To ensure the pattern of connectivity we observed between groups (decreased right auditory cortex and right insula to L-DLPFC connectivity) was not due to intrinsic, task-irrelevant differences in neural architecture, the data from the language task was compared to that collected during passive listening to tone sequences. Whereas during speech perception, amusic subjects showed reduced functional connectivity between left frontal and right insula/auditory ROIs relative to controls (p=0.0001 for both ROIs; in line with the whole-brain imaging analyses), this pattern did not hold during passive listening to tones (Amusia vs. Control connectivity, p = 0.29, Group (Amusic, Control) by Task (Speech Perception, Passive Tone Listening) interaction p = 0.045 for the insula ROI; Amusia vs. Control p = 0.30, Group by Task interaction p = 0.035 for the auditory cortex ROI – see Figure 6). These interactions suggest that our neural connectivity results are specifically linked to speech perception, rather than reflecting an overall connectivity difference between groups regardless of task state.”

Reviewer #2:[…]The study is well-conducted and analyzed, and the paper is well-written. But I am not persuaded about the interpretation of the results. The main problem is the lack of any explicit brain-behavior relationship. This issue takes two forms. First, the brain data and the behavioral data are never related to one another. The authors make the claim that the functional connectivity differences are related to the behavior, but this is not shown by any analysis. Without any such link, it is hard to know whether the connectivity differences observed have anything to do with the particular ability sampled by this task, or if they are related to something else entirely.

Thanks to you and reviewer 1 for bringing this up – we have taken all of your suggestions to heart and run the analyses. To avoid adding yet more text, please see reviewer 1 point 1.

Second, the behavior itself does not differ across participant groups. The authors specifically desired this outcome so that any imaging differences would not be driven by group behavioral differences. But that idea does not make sense to me because there were two conditions, one using pitch cues and another using duration cues. Presumably, amusic persons have trouble with the former but not the latter. Yet this was not the case in the behavior (no condition X group interaction), nor in the imaging data (the FC effects were not influenced by condition either). So if the FC results are somehow related to the downweighting of pitch cues it should show up specifically in the condition where such cues were available, and not in the condition where duration cues were available. Without this crucial dissociation, the interpretation does not hold together.

Please see our responses to the second and third editorial comments.

In the Discussion section the authors argue that the deficit in amusia has more to do with tonal retention than perception, an idea which is supported by other authors, although the bulk of the evidence suggests that both features are present. But it is not entirely clear why a retention deficit would be so important in the task used, which does not appear to require a lot of working memory since the spoken stimuli simply have to be matched to a visual representation of the sentence.

We would argue that the present task can be suggested to have a non-trivial ‘working memory’ component. Each auditory stimulus was about 2.5 s long and participants needed to retain both of them in memory while comparing them to the visual stimulus. Moreover, memory is inextricably involved with prosody perception because acoustic cues to prosodic features are compared to the preceding context. With regards to pitch, for example, pitch accents and phrase boundaries are conveyed via a change relative to the contextual baseline F0. Of course, this is somewhat true for segmental speech perception as well, inasmuch as the acoustic cues associated with phonemes are modulated by coarticulation. However, the time scale over which these contextual effects take place is much longer in suprasegmental than in segmental perception.

Another related issue is that in amusia there is often a wide range of perceptual performance, with some individuals scoring near chance on pitch discrimination and others scoring at the same level as non-amusic persons. It's not clear how that heterogeneity may have influenced performance on this task, especially as the pitch intervals used in the stimuli are not documented in the manuscript, which seems like a significant omission. It would be important to know whether the excursions in F0 for the pitch-varying sentences (which are typically much larger than one or two semitones) were smaller than the thresholds that the amusics can perceive or not. In general, there is no discussion of individual differences in the data, which might have helped to understand the findings.

Thank you for pointing out the omission; we now supply details regarding the size of the F0 differences between stimulus pairs in the prosody stimuli. In every case, these differences were larger than the pitch discrimination threshold of even the ‘most impaired’ amusic participant. We note this in the Discussion section.

[Editors’ note: further revisions were suggested prior to acceptance, as described below.]

Essential revisions:Brain-behavior correlations. The inclusion of the current Figure 5 certainly strengthens the manuscript and is an important addition. However, because the correlation is driven primarily by the between-group differences, the importance of the overall correlation should not be overstated. In particular, it is probably not appropriate to mention it in the abstract without the necessary caveats. Instead just a statement, saying that the between-group differences in connectivity reflect the between-group differences in performance in the same groups of listeners, would suffice.

Thanks for raising this. To avoid overstating the importance of the brain-behavior correlations, we have removed mention of them in the Abstract and included, as suggested, a statement saying that the between-group differences in connectivity reflect the between-group differences in performance in the same groups of listeners.

“[..] reflected the between-group differences in cue weights in the same groups of listeners.”

Behavior task. There are still some questions remaining on this front. In particularly, there remain some questions as to why the amusics should show the same amount of behavioral gain when pitch cues are added. Is this thought to be because the pitch changes are so large as to influence performance despite impaired pitch processing? A little more discussion of this aspect of the results will likely help readers follow the logic better. There is also some skepticism regarding when and whether participants knew which cues were informative on each trial (Discussion section). For instance, in many cases participants would probably have been able to complete the task following just the first interval, rather than have to wait for the second interval and make a comparison. Although the 2AFC paradigm makes the analysis of performance easier, it cannot be assumed that participants always make full use (or need to make use) of the information from both intervals. This observation has some implications for the extent to which memory processes are involved.

First, so that readers can experience the task themselves, we have created a simulation of the in-scanner task in Gorilla Open Materials so that readers may find that – as we do – the task is not particularly easy even in quiet listening conditions (https://gorilla.sc/openmaterials/102786). We have additionally included this link to the task in the manuscript in the Data Availability Statement. Second, we have analyzed reaction times in the task, measured in quiet listening conditions outside the scanner in the same participants and found that they were quite long, with an average of 1.64 seconds after the end of the second auditory presentation. This suggests that participants did not respond immediately, but needed time to mentally compare the two auditory presentations in memory in order to make their judgments. We have added this text to the Discussion section accordingly:

“Our task arguably taxed working memory resources: in a similar paradigm performed by the same participants in quiet listening conditions (Jasmin et al., 2020a), the mean reaction time measured from the end of the second auditory stimulus was 1.64 seconds, indicating that participants needed some time to compare both auditory presentations and make their judgments.”

Additional analysis. There are two areas where additional analyses may be illuminated. First, the main conclusions of the study rely in part on the results of the correlations between pitch cue weights and connectivity measures. Duration cue weights, which do not differ between groups, were also measured. It appears important, as a negative control, to comment on the absence (or not) of correlations between these duration cue weights and connectivity measures (subsection “Correlations between functional connectivity levels and prosodic cue weights”).

Thank you for suggesting this analysis. We should have referred to the cue weights more consistently in the manuscript to avoid confusion. The ‘normalized prosodic cue weights’ used in the paper are relative weights that reflect the degree to which participants relied upon pitch versus duration. They are normalized with respect to one another, such that possible values range from 0 to 1, with values greater than 0.5 indicating greater reliance on pitch than duration, and values less than 0.5 indicate greater reliance on duration than pitch. So-called ‘raw’ pitch and duration cue weights, regression coefficients used to calculate relative normalized weights, are not easily interpretable in isolation because the raw weights can be influenced by factors such as the general degree of attention to the task. For this reason, it is not possible to perform the suggested control analysis, but to address the issue of confusing language we have changed all instances of ‘pitch cue weight’ to ‘normalized cue weights’ in both the main text and the figure axis labels, and clarified as follows in the main text:

“Of the 30 participants in this study, 21 took part in an experiment that measured the degree to which they relied on pitch versus duration to categorize prosody, i.e. their ‘normalized prosodic cue weights’, which ranged from 0 to 1, with values greater than 0.5 indicating greater reliance on pitch than duration, and values less than 0.5 indicate greater reliance on duration than pitch (Experiment 1, Jasmin, et al., 2020a).”

Second, the comparison of connectivity between the active and passive (tones) task was based on the ROIs derived from the passive task, with the assumption that the passive task should not result in differences in connectivity. However, as far as we could tell, the initial data-driven connectivity tests were not run separately on the passive task. If that were done just on the passive task, would any connectivity differences between the groups emerge? If not, that would provide a further important negative control.

This is a good point: indeed only the passive task (tone) data were analyzed with respect to ROIs generated from the active task. As you suggest, we have run an analogous version of the first step of the data-driven connectivity analysis on the passive task data. No group differences in global connectivity were detected at the threshold used in the paper (P<4.61) nor indeed even at an uncorrected threshold of p<.001. We now have added this to the text, thank you for the suggestion.

“An analogous procedure was run on the passive tone listening data, in which whole-brain connectedness values were compared by Group (amusic vs. control) in a linear mixed effects model. No significant FDR-corrected group differences were detected, nor at a reasonable uncorrected threshold of p<.001.”